# Vision-Free Object 6D Pose Estimation for In-Hand Manipulation via Multi-Modal Haptic Attention

**Chanyoung Ahn**[1]    **Sungwoo Park**[1,2]    **Donghyun Hwang**[1]

[1] Korea Institute of Science and Technology    [2] Korea University
{chanyoung.ahn, sungwoo.park, donghyun}@kist.re.kr

**Abstract:** Humans are capable of in-hand manipulation without visual feedback, by inferring its pose through haptic feedback. However, in-hand manipulation with multi-fingered robotic hands remains highly challenging due to severe self-occlusion and limited visual accessibility. To address this problem, we propose a vision-free approach that integrates multiple haptic sensing modalities. Specifically, we develop a haptic attention-based pose estimator that captures correlations among kinesthetic, contact, and proprioceptive signals, as well as their temporal dynamics. Experimental results demonstrate that haptic feedback alone enables reliable pose estimation and that contact-rich sensing substantially improves reorientation performance. Our pose estimator achieves average errors of only 4.94 mm in position and 11.6 degrees in orientation during 300 iterations (10 seconds), underscoring the effectiveness of haptic-driven pose estimation for dexterous manipulation. Videos are available at https://cold-young.github.io/haptic-estimator/

**Keywords:** Haptic Feedback, In-Hand Manipulation, Dexterous Manipulation

## 1   Introduction

Humans can perform in-hand manipulation, such as rotating a ball without visual feedback, by estimating pose from haptic sensations. For robots, pose information is equally critical, yet in-hand manipulation is more challenging than quasi-static tasks due to frequent contacts and occlusions. While recent work, including that of OpenAI [1], has advanced dexterous manipulation, most methods remain vision-centric [2]. However, vision alone struggles with occlusion and unobservable object states, often requiring multiple cameras and complex setups that are impractical for embodied systems.

Previous approaches have mainly relied on visual feedback, with haptics—especially tactile—receiving less focus [3, 4, 5]. Fingertip tactile sensors capture slip or re-grasping but provide only local information, while kinesthetic sensing from joints has been applied to recognition or simple rotations [6, 7, 8]. Yet, these modalities are rarely integrated, leaving open the challenge of a unified representation that exploits their complementary nature.

To address this gap, we propose an attention-based multi-modal haptic pose estimator that integrates proprioceptive, kinesthetic, and cutaneous signals across time. Attention adaptively weights informative features, enabling reliable estimation from rich haptic input alone. Our method achieves 4.94 mm position and 11.6 degrees orientation error over 300 iterations, and maintains stable reorientation for 27 seconds without dropping the object. These results demonstrate the feasibility of vision-free haptic intelligence for dexterous manipulation and highlight its potential for real-world applications.

The principal contributions of this work are as follows.

Presented at the 2nd Workshop on Dexterous Manipulation at CoRL 2025, Seoul, South Korea.

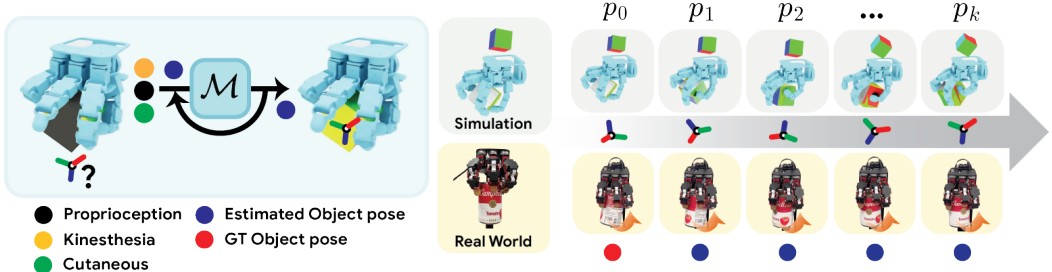

Figure 1: **Vision-free object pose estimation.** *(Left)* A multi-fingered hand with three haptic modalities interacts with the object. *(Right)* The model recurrently predicts the pose $\hat{p}_k$ from haptic history, the initial pose $p_0$, and the previous estimate $\hat{p}_{k-1}$.

1. **Attention-based multi-modal haptic pose estimation**: We propose an LSTM attention-based estimator that integrates kinesthetic, cutaneous, and proprioceptive signals, enabling vision-free object pose estimation.

2. **Empirical validation in simulation and real world**: We demonstrate that robots can reliably estimate and track object pose purely from haptic feedback, achieving accurate reorientation for up to 27 seconds without failure.

3. **Highlight the utility of kinesthetic sensing**: We show that kinesthetic feedback from robot joints, an underexplored sensing modality, provides valuable cues that significantly enhance pose estimation in dexterous manipulation.

## 2 Vision-Free Object Pose Estimator

The goal of our approach is to achieve vision-free object pose estimation for dexterous in-hand manipulation, where visual sensing is often unreliable due to occlusion or limited accessibility. The central challenge lies in inferring accurate object pose purely from haptic feedback, which requires integrating multi-modal and temporally extended signals.

### 2.1 Preliminary

**Task & Skill Policy.** We consider the in-hand object reorientation task introduced in [2], which requires accurate and continuous object pose estimation during mid-air manipulation without dropping the object. To collect demonstration data, we first train a skill policy to stably grasp an object lifted from the table. Using 1024 simulated instances, we initialize stable grasp configurations and apply curriculum learning by gradually increasing gravity and target orientation difficulty. Once the policy converges to an expert level, it is used to generate trajectories for training and evaluating the proposed pose estimator. This setup ensures that our estimator is validated under realistic reorientation conditions where precise pose feedback is essential.

**Observations and Problem Formulation.** At each timestep $t$, the robot receives multi-modal haptic observations: proprioception, consisting of joint angles $\mathbf{q} \in R^{16}$; kinesthetic sensing, consisting of forces/torques measured at the base of each finger joint $\mathbf{t} \in R^{12}$; and cutaneous sensing, consisting of binary fingertip contact signals $\mathbf{c} \in R^4$. We denote the aggregated haptic history from the previous 32 steps as $\mathbf{S}_{t-32:t-1} = \{\mathbf{q}, \mathbf{t}, \mathbf{c}\}$. The estimator also takes the prior pose estimate $\hat{p}_{t-1}$. The objective is to predict the pose of the current object $\hat{p}_t$, defined as position and orientation in $SO(3)$. Formally, the estimator is given by: $\hat{p}_t = \mathcal{M}(\hat{p}_{t-1}, \mathbf{S}_{t-32:t-1})$.

**Model Architecture.** Our vision-free object pose estimator is illustrated in Figure 2. A bidirectional LSTM encoder is used to capture long-horizon temporal dependencies in the multi-modal haptic sequence. On top of the encoder, we employ additive attention to adaptively weight informative timestep and modalities. To further emphasize tactile evidence, we introduce a contact bias, which

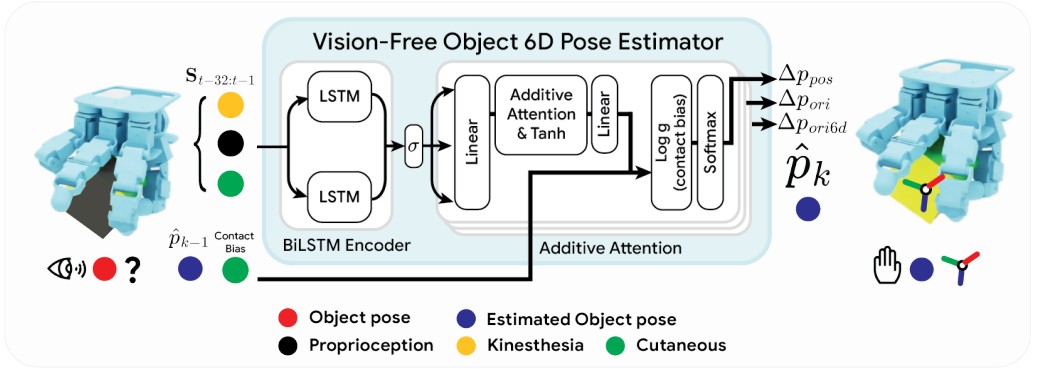

Figure 2: **Proposed BiLSTM-based haptic pose estimator.** multi-modal haptic histories $\mathbf{S}_{t-32:t-1}$ and the previous estimate are used to predict the current object pose $\hat{p}_k$.

increases the attention weights at contact events and reduces the contribution of signals when no contact occurs. The regression head predicts differential pose updates in both position ($\Delta p_{pos}$) and orientation ($\Delta p_{ori}$). These values are recursively integrated with the previous estimate $\hat{p}_{t-1}$ to obtain the current object pose $\hat{p}_t$. This design allows the model to incorporate long-horizon haptic dependencies while adapting to fine-grained contact cues essential for in-hand manipulation.

# 3 Experiments

Our goal is to evaluate whether multiple haptic sensing modalities can enable reliable object pose estimation for in-hand reorientation tasks. We use three types of haptic signals: proprioceptive, kinesthetic, and skin detection as the only sources of information for the estimation of poses. More detail information in Section A.

## 3.1 Simulation: Pose Estimation Accuracy

We first evaluate the accuracy of our pose estimator in simulation while an expert policy performs in-hand reorientation tasks. During each rollout, the model receives haptic observations and recursively predicts the object pose, which is compared against ground-truth pose from the simulator. We measure average position and orientation errors accumulated over 300 steps. The proposed estimator achieves 4.94 mm mean position error and 11.6 degrees mean orientation error. Figure 6 illustrates the estimated results over time.

Interestingly, error accumulation did not grow linearly with time. Larger orientation errors tended to occur when the object experienced abrupt orientation changes, suggesting that dynamic transitions are more challenging for haptic-only estimation.

## 3.2 Simulation: Task Performance with Pose Feedback

We evaluate our approach in the in-hand reorientation task, running each experiment for 3,600 steps (120 seconds). Performance is assessed with two metrics:

- **Time to Terminate (TTT)**: The elapsed time until the object becomes stuck, is dropped, or deviates from the target axis.
- **Target Success**: The number of successful reaches to the target orientation.

We compare two conditions: (1) a ground-truth oracle baseline using privileged object pose, and (2) our proposed haptic-only estimator. Each condition is tested with three random seeds. As summarized in Table 1, the oracle baseline achieves an average stability of 88.7 seconds and 77.3 successful reorientations, while our estimator maintains stability for 27.1 seconds with 3.3 successful reorientations.

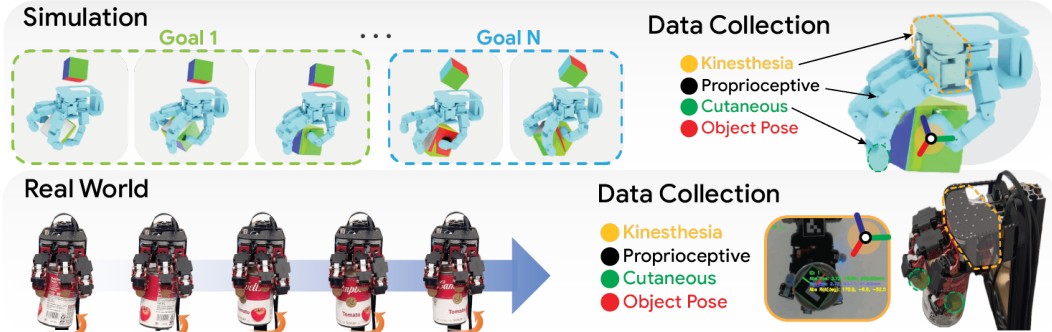

Figure 3: **Manipulation tasks and data collection in simulation and the real world.** *(Top, Simulation)* The robot hand reorients objects toward multiple goal orientations within a fixed horizon. *(Bottom, Real World)* A cylindrical object is continuously rotated about its axis using the same policy.

Table 1: Evaluation of object reorientation performance.

| Method | TTT [s] | Target Success [#] |
|---|---|---|
| Ground Truth (GT) | $88.7 \pm 29.2$ | $77.3 \pm 27.2$ |
| No Info. (Random) | $1.42 \pm 0.1$ | $0.0$ |
| **Ours (Haptic-only)** | $27.1 \pm 23.1$ | $3.3 \pm 3.2$ |

These results demonstrate the feasibility of vision-free haptic-based pose estimation, while also highlighting the performance gap relative to ground-truth sensing. Importantly, they underline the critical role of accurate real-time pose estimation: stable grasping can be achieved without precise feedback, but successful reorientation requires fine-grained and continuous pose tracking.

### 3.3 Real-World Validation: In-Hand Rotation

Finally, we evaluate the feasibility of our method on a physical platform. Using a canned tomato object with an AprilTag attached to its bottom surface (Figure 6), we collect 20K haptic samples and ground-truth poses while executing an in-hand rotation policy. Haptic feedback is recorded directly from the robotic hand platform, and details of the setup are provided in Section A.

We then train the model solely on this real-world dataset, which is nearly 200 times smaller than the simulated dataset. Over 10 seconds (100 steps) of continuous inference, the estimator achieves an average error of 38.2 mm in position and 3.67 degrees in orientation. Although performance is lower than in simulation, the results demonstrate feasibility and emphasize the need for larger and more diverse real-world datasets to close the sim-to-real gap.

## 4   Conclusion and Limitation

The ability to manipulate objects in free space using only touch remains a central challenge in robotics. We proposed a vision-free pose estimator that integrates proprioceptive, kinesthetic, and cutaneous signals, showing that haptic feedback alone enables reliable pose estimation and improves reorientation performance. These results suggest that robots, like humans, can reason about objects without vision.

Our study is limited to simple reorientation tasks and a narrow set of object geometries. Scaling to more complex tasks and broader categories will require richer data and model refinement. Future work will also explore additional sensing (e.g., finger joint torque) and improved modeling of spatial sensor relationships to enhance generalizability.

**Acknowledgments**

This work was supported in part by the Industrial Strategic Technology Development Program (RS-2024-00442029) funded by the Ministry of Trade, Industry and Energy (MOTIE, Korea), and in part by the KIST Institutional Program. (Corresponding author: Donghyun Hwang)

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

# Appendix

## A  Experimental Setup

**Simulation Setup.** We design a learning environment in Isaac Lab [9] to train our pose estimator, skill policy. The simulation frequency is 120Hz, and the control frequency is 30Hz.

**Hardware Setup.** We use a four-fingered robotic hand platform [10]. The robotic hand platform is capable of kinesthetic feedback by utilizing a three-axis F/T sensor on each finger as shown in Figure 4 It weighs approximately 550 g and has a maximum payload capacity of around 3 kg. The detailed specifications of the robotic hand platform and three-axis F/T sensor for kinesthetic feedback. Each joint of the robotic hand is controlled by a PD position controller with a control frequency of 1kHz. The target position commands are converted to torque using a PD controller ($K_p = 1.0$, $K_d = 0.1$).

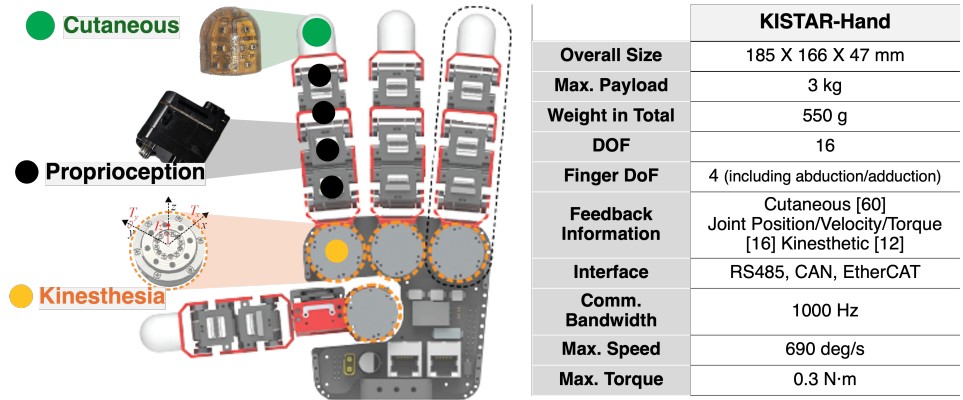

| | KISTAR-Hand |
| --- | --- |
| **Overall Size** | 185 X 166 X 47 mm |
| **Max. Payload** | 3 kg |
| **Weight in Total** | 550 g |
| **DOF** | 16 |
| **Finger DoF** | 4 (including abduction/adduction) |
| **Feedback Information** | Cutaneous [60] Joint Position/Velocity/Torque [16] Kinesthetic [12] |
| **Interface** | RS485, CAN, EtherCAT |
| **Comm. Bandwidth** | 1000 Hz |
| **Max. Speed** | 690 deg/s |
| **Max. Torque** | 0.3 N·m |

Figure 4: Illustration of robotic hand platform capable of multi-modal haptic feedback at each finger. *(left)* show the three-axis F/T sensor embedded into the robotic hand platform for kinesthetic feedback, the robotic hand platform, and the robotic finger, respectively. *(right)* Specifications of robotic hand.

## B  Training Procedure

We train our control policy using the proximal policy optimization (PPO) algorithm [11] with a multilayer perceptron (MLP) for both the policy and value networks (Figure 5). For simulation, we use Isaac Sim with Isaac Lab [9], setting the timestep to $dt = 1/120$ s with four simulation substeps. The simulation runs 8192 parallel environments, and the policy executes actions every four steps, corresponding to a 30 Hz control frequency. Policies are trained for 100K steps using the skrl RL library [12]. All experiments are conducted on single RTX 4090 GPU.

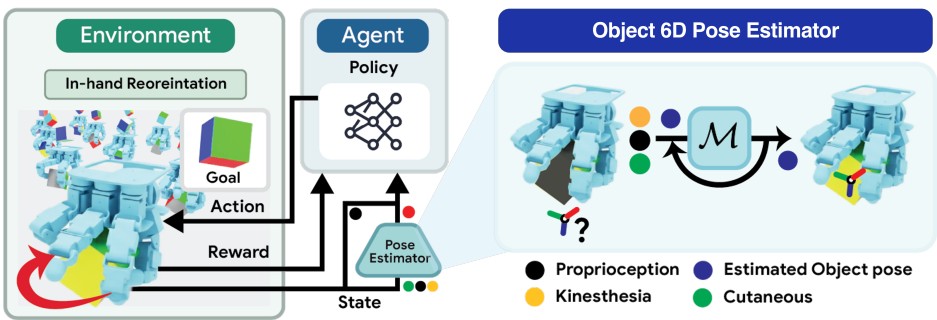

Figure 5: Overall Learning System Architecture. To manipulate an object without oracle object information, the robotic agent receives estimated object 6D pose from our estimator.

## C  Vision-Free Object Pose Estimator

See in Figure 6. From top to bottom, the plots show the ground-truth (GT) and predicted values for the $x$, $y$, and $z$ positions, respectively. The bottom plot illustrates the orientation error. Predictions are generated in an autoregressive manner, where each estimated pose is recursively fed back into the model as input for the next prediction. The overall trajectories in Figure 7.

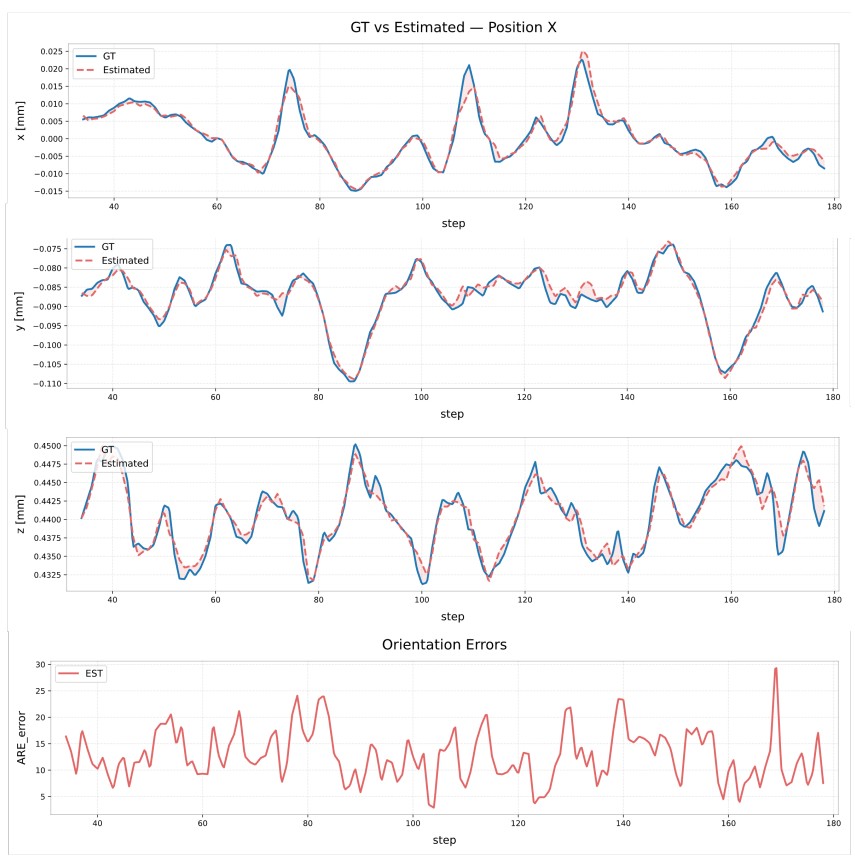

Figure 6: Object pose estimation results in the in-hand reorientation task (simulation).

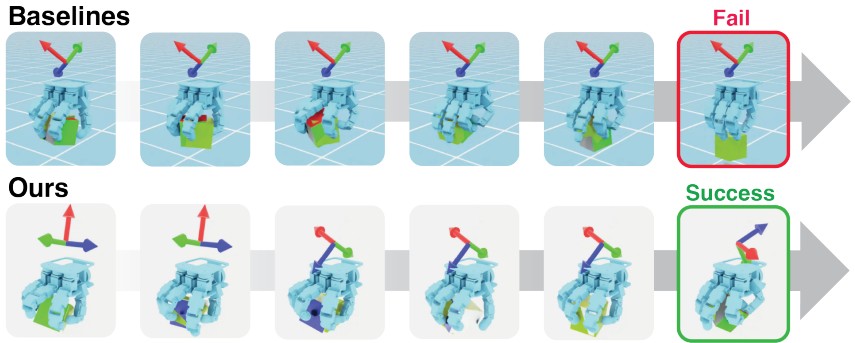

Figure 7: Evaluation Trajectories for object reorientation performance.

