# OpenReview forum: "Vision-Free Object 6D Pose Estimation for In-Hand Manipulation via Multi-Modal Haptic Attention"
_robot-learning.org/CoRL/2025/Workshop/Dexterous_Manipulation — CoRL 2025 Workshop Dexterous Manipulation Spotlight_

### Official Review · Reviewer_WtZX · 2025-09-08
**Accept**

**Rating:** 7
**Confidence:** 5

**Review:**

The paper proposes a vision-free pose estimation framework for in-hand manipulation using multimodal haptic signals (proprioceptive, kinesthetic, and cutaneous). An LSTM-based attention model is developed to integrate temporal haptic data for accurate pose prediction. Experiments in simulation and on a physical robotic hand show that the method achieves promising accuracy in estimating object pose and maintaining stable reorientation without visual input.

Strengths:
* The paper clearly identifies the limitations of vision-based methods for dexterous manipulation and presents a well-motivated and well-written solution
* Combining multiple haptic sources to infer object pose is interesting and highlights the complementary role of kinesthetic sensing, which is often underexplored.
* The authors conduct thorough evaluations in both simulation and the real world, showing great performance.

Weaknesses:
* The method requires extensive real-world trajectory data for training, which may be difficult to collect at scale, raising concerns about practicality
* Experiments are limited to a set of fixed objects, leaving open whether the approach can generalize effectively to unseen objects

---

### Official Review · Reviewer_eQoC · 2025-09-09

**Rating:** 7
**Confidence:** 5

**Review:**

Summary: This paper proposes a vision-free approach to object pose estimation for in-hand manipulation using a multi-modal haptic attention model. The method integrates proprioceptive, kinesthetic, and cutaneous signals with a BiLSTM + attention framework. Experiments in both simulation and real-world settings demonstrate feasibility, achieving 4.94 mm / 11.6° average error in simulation and showing sim-to-real applicability.

Strengths:
* Addresses an important problem: pose estimation without vision, highly relevant to dexterous manipulation where occlusion is inevitable.
* Proposes a clear multimodal haptic attention mechanism, emphasizing underexplored kinesthetic feedback.
* Provides both simulation and real-world validation.

Weaknesses / Concerns:
* Task evaluation is limited to relatively simple reorientation with narrow object geometries. Generalization to more complex shapes remains untested.
* The control performance (27 s vs 88 s for oracle) and success shows that the approach is promising but still far from practical deployment.

This work is highly relevant to CoRL-Dex-2025, as it focuses on haptics and in-hand manipulation. The work contributes to ongoing conversations on multimodal sensing and dexterous manipulation. While the results are modest compared to vision-based baselines, exploring vision-free haptic pose estimation is meaningful and fits well within the scope of the workshop. I recommend acceptance.

---

### Decision · Program_Chairs · 2025-09-18

Accept (Spotlight)